# Baseline Radiomics as a Prognostic Tool for Clinical Benefit from Immune Checkpoint Inhibition in Inoperable NSCLC Without Activating Mutations

**DOI:** 10.3390/cancers17111790

**Published:** 2025-05-27

**Authors:** Fedor Moiseenko, Marko Radulovic, Nadezhda Tsvetkova, Vera Chernobrivceva, Albina Gabina, Any Oganesian, Maria Makarkina, Ekaterina Elsakova, Maria Krasavina, Daria Barsova, Elizaveta Artemeva, Valeria Khenshtein, Natalia Levchenko, Viacheslav Chubenko, Vitaliy Egorenkov, Nikita Volkov, Alexei Bogdanov, Vladimir Moiseyenko

**Affiliations:** 1N.P Napalkov Saint Petersburg Clinical Research and Practical Centre for Specialized Types of Medical Care (Oncological), Leningradskaya Str. 68A, 197758 Saint Petersburg, Russia; moiseenkofv@gmail.com (F.M.); nadya.cvetkova@mail.ru (N.T.); veracernobrivceva@gmail.com (V.C.); albina_zhabina@inbox.ru (A.G.); ani101192@mail.ru (A.O.); stepanova100992@mail.ru (M.M.); ekostepanova@gmail.com (E.E.); mary-krasavina2011@yandex.ru (M.K.); dasha.barsova@yandex.ru (D.B.); mukhina_ev@mail.ru (E.A.); heinstein@mail.ru (V.K.); levch.nv@gmail.com (N.L.); vchubenko@me.com (V.C.); v.egorenkov@inbox.ru (V.E.); volkovnm@gmail.com (N.V.); aleks_aa@mail.ru (A.B.); moiseyenkov@gmail.com (V.M.); 2N.N. Petrov National Medical Research Center of Oncology, Ministry of Public Health of the Russian Federation, Leningradskaya Str. 68, 197758 Saint Petersburg, Russia; 3Department of Experimental Oncology, Institute of Oncology and Radiology of Serbia, 11000 Belgrade, Serbia; 4Department of Oncology, Pediatric Oncology and Radiation Therapy, St.-Petersburg State Pediatric Medical University, St. Lithuanian 2, 194100 Saint Petersburg, Russia

**Keywords:** NSCLC, prediction, prognosis, radiomics, machine learning, ensemble, immunotherapy, checkpoint inhibitors

## Abstract

This study introduces a powerful machine learning-based radiomics approach to help improve predictions of immunotherapy outcomes in patients with non-small cell lung cancer (NSCLC). We believed that the full potential of CT scan-based tumor analysis had not been achieved, partly due to limited use of model integrations (ensembles) in previous research. To address this, we tested 1680 combinations of data processing and machine learning methods, selecting the best-performing ones to create an integrated (ensemble) model. Using clinical and imaging data, our final model achieved an AUC of 0.86 for predicting 24-month patient survival, which, to our knowledge, exceeds previously published results for this diagnosis and disease outcomes. This approach reduces the weaknesses of relying on a single model and offers a more reliable and accurate tool for predicting immunotherapy outcomes.

## 1. Introduction

Non-small cell lung cancer (NSCLC) is a common, deadly, and highly heterogeneous tumor type. This heterogeneity is evident in its clinical course, histological characteristics, genetic and expression profiles, and in responses to treatment. Different treatment modalities can lead to diverse outcomes; for example, some therapies provide immediate tumor shrinkage and symptom improvement, while others deliver a survival advantage over the long term [1]. The combination of these benefits forms the basis for treatment decisions in clinical practice.

Historically, chemotherapy was the main NSCLC treatment, often resulting in significant tumor shrinkage, though typically for only a short duration. Targeted therapies, on the other hand, have shown the potential for longer-lasting responses but are limited to a small group of patients with specific molecular aberrations. More recently, immunotherapy, particularly checkpoint inhibitors, has emerged as a transformative treatment, offering remarkably long survival benefits for a subset of patients [2].

The selection of patients who are likely to benefit from immunotherapy is based on several factors, including the carcinoma’s origin (with smoking history playing a role), the molecular genetic profile (such as mutations associated with sensitivity or resistance and tumor mutational burden), and signs of immune system activation. Among these factors, PD-L1 expression is currently the primary criterion for estimating both the likelihood of an overall response and a long-lasting response. PD-L1 also serves as a marker of the mechanism by which tumor cells protect themselves from activated immune cells, highlighting the complex interactions among the tumor, its microenvironment, and the host organism. However, PD-L1 status has notable limitations; for instance, approximately 20% of patients with high PD-L1 expression still experience early progression, while a proportion of PD-L1-negative patients do respond to treatment. These challenges underscore the need for more refined patient-selection strategies, with the goal of improving outcomes and personalizing treatments in the heterogeneous landscape of NSCLC.

The selection of NSCLC patients who are likely to benefit from immunotherapy is increasingly approached by use of CT radiomics, which reveals subtle intra-tumor heterogeneity that cannot be identified by visual inspection of imaging scans. Radiomics quantitatively characterizes tumor imaging data by extracting features that capture tumor size, shape, voxel intensity distribution, spatial relationships, and texture patterns within the tumor VOI [3]. These features are then entered into machine learning models to classify patients according to their clinical outcomes and therapy response [4].

Current state-of-the-art approaches for classifying therapy response and disease outcomes in NSCLC still leave considerable room for improvement [5]. Radiomics still achieves only moderate predictive power on held-out test sets, with reported AUCs rarely exceeding 0.75. Consequently, radiomics only has potential to become a supportive biomarker rather than a stand-alone decision tool. Progress is limited by the absence of optimized, standardized, and widely adopted pipelines for feature selection and classification only sporadic use of ensemble techniques and limited reproducibility due to rare sharing of code, hyperparameters, and imaging protocols. Until sequential internal validation and external validation become routine, radiomics will remain confined to proof-of-concept research instead of becoming a reliable clinical tool [6,7,8,9,10,11,12,13]. To address these gaps, we aimed to improve the consistency and predictive accuracy of radiomics-based classification by employing an ensemble approach that integrates the best-performing individual models. Such an ensemble radiomics framework mitigates the limitations of single-model variability and improves overall robustness and generalizability, providing a foundation for more reproducible and accurate outcome prediction in future studies.

The novelty of this study is based on systematic evaluation of 1680 standardized pipelines combining normalization, preprocessing, feature selection, and classification steps. It also introduces an ensemble framework that integrates top-performing models, to reduce reliance on any single model’s chance success and enhance predictive stability. Importantly, ensembling was performed by soft voting, averaging continuous probability scores of individual models to avoid the bias of hard voting in selecting probability score thresholds. Several previous ensemble studies in NSCLC used XGBoost classifier alone [14,15], whereas we used a similar AdaBoost classifier alongside nine others. One study ensembled five classifiers using hard voting using a fixed 0.5 probability score threshold for each individual model [16]. The most comprehensive prior approach combined RF, SVM, and LASSO to build 54 predictive models [8]. Notably, many studies use the term “ensemble” to describe the integration of multiple feature types rather than distinct radiomic workflows [8,15,17]. Our current study thus presents a substantial methodological advance by addressing workflow optimization and standardization through an exhaustive pipeline, supported by openly shared code for reproducibility and wider adoption.

Motivated by the clinical need for more reliable prognostic tools in immunotherapy-treated NSCLC patients, we aimed to implement a comprehensive workflow to systematically explore 1680 combinations of normalization, preprocessing, feature selection, and classification strategies on pretreatment CT scans. The top-performing models were then integrated into a unified ensemble radiomics signature, aiming to enhance generalizability, reduce variance due to chance performance of individual models, and support more robust prognostic assessments.

## 2. Materials and Methods

### 2.1. Patient Group

We used a retrospective database of patients treated for inoperable NSCLC at the N.P Napalkov Saint Petersburg Clinical Research and Practical Centre for Specialized Types of Medical Care (Oncological) in 2021. This study was approved by the institutional Ethics Committee (Approval No. 3, dated 14 March 2023). Patients received checkpoint inhibitors (pembrolizumab and bioanalogues, atezolizumab, nivolumab, or prolgolimab) as first-line palliative therapy, either as monotherapy or in combination.

Only patients with available pre-immunotherapy images were included. Outcomes were defined as 6- or 12-month progression-free survival (PFS) and 24-month overall survival (OS). Clinical characteristics are presented in Table 1. All the patients were treated according to national guidelines and MDT decisions, receiving a checkpoint inhibitor either alone or with chemotherapy as part of routine care. Treatment response was evaluated with CT every 6–8 weeks, adverse events were monitored per local practice, and overall survival was determined via patient phone contact or the national health database. Progression of disease was defined according to iRECIST criteria. Progression of disease was considered as a time from day one of first cycle to date of first registered progression, provided that this progression was subsequently confirmed by a second investigation.

The prospective sample size calculation was based on a pilot study of the first 120 chronologically included patients, which indicated that a minimum of 96 patients, including 12 positive cases, would be required. The calculations were based on an alpha value of 0.05, a beta value of 0.20, a positive-to-negative case ratio of 0.14, and an expected effect size corresponding to an AUC of 0.75. The final study included 220 patients, with at least 147, 75, and 37 survivors (positive cases) observed in the 6-, 12-, and 24-month groups, respectively. All actual parameters exceeded the initial sample size estimates, with positive-to-negative ratios of 0.67, 0.34, and 0.17, respectively, and an achieved highest prognostic AUC of 0.86.

### 2.2. Image Acquisition

CT chest images from 220 patients were acquired in uncompressed DICOM format (slice thickness ≤ 2.5 mm) using a Siemens Somatom Definition 128 CT scanner (Siemens, Munich, Germany) at the N.P. Napalkov Saint Petersburg Clinical Research and Practical Centre for Specialized Types of Medical Care (Oncological). The dataset excluded low-dose protocols, while the use of contrast was not a limiting factor. During database creation, the radiologist was responsible for data selection, interpretation, and primary reporting, image labeling, and data deidentification. CT scanning and preliminary analysis were performed using a RadiAnt DICOM Viewer v2025.1. Data processing involved preparing the CT datasets, image pre-processing, detecting pathological lesions, and segmentation. Contouring of pathological lung lesions was performed in 3D Slicer v.5.8.1, an open-source medical image analysis platform, using multiplanar reconstruction (MPR) of chest CT images. This process was guided by initial radiology reports to ensure accuracy and consistency. Tumor VOIs were semi-automatically segmented slice-by-slice in the axial plane using the “Grow from Seeds” tool (FastGrowCut) in 3D Slicer’s Segment Editor by a radiation oncologist (F.M.) with 15 years of oncologic imaging experience. Threshold-based selection, paint, erase, and island removal tools were used to refine segmentations. The final tumor VOIs were reviewed for consistency and exported in MRB format for further analysis. Figure 1 illustrates the final segmentation.

### 2.3. Feature Extraction

For the radiomics analysis, we used the open-source Python package Pyradiomics v3.1.0, which is compliant with the Imaging Biomarker Standardization Initiative (IBSI) standards. [18]. Using the parameter file provided below, the software was configured to generate all available image transformations and feature types, resulting in a total of 2157 features computed per CT scan. Seven image transformations were applied wavelet, square, square root, logarithm, gradient, exponential, and Laplacian of Gaussian (LoG).

To reduce inter-scan variability, the CT scans were first z-score normalized (mean intensity = 0, standard deviation = 1) and then resampled to an isotropic voxel size of 1 × 1 × 1 mm using ‘sitkBSpline.’ Radiomic features were extracted only from tumor VOIs, yielding 107 standard shape, intensity (first-order), and texture (second-order) features from the original images, while higher-order features were computed from the transformed images.

The bin width was individually determined for each filter type in a pilot analysis of 75 CTs to keep the number of gray-level bins per scan between 30 and 130, which is considered optimal for textural reproducibility without causing over-smoothing or excessive noise sensitivity [19]. This was necessary because the seven image filter transformations produced images with distinct pixel intensity ranges, requiring separate bin-width settings for each filter to maintain a consistent bin count across all individual scans. For detailed descriptions of the extracted radiomic features, please refer to https://pyradiomics.readthedocs.io/en/latest/features.html (accessed on 25 May 2025).

Below is the Params.yaml file:
setting:
    normalize: true
    normalizeScale: 600
    resampledPixelSpacing: [1, 1, 1]
    interpolator: 'sitkBSpline'
    voxelArrayShift: 1000
    binWidth: 30.0
    label: 2
imageType:
    Original: 
    LoG: 
        sigma: [1.0, 2.0, 3.0, 4.0, 5.0]
        binWidth: 15.0
    Wavelet: 
        binWidth: 8.0
    Square: 
        binWidth: 15 
    SquareRoot: 
       binWidth: 25
    Logarithm:
        binWidth: 50
    Exponential:
        binWidth: 6
    Gradient: 
        binWidth: 14
featureClass:
    glcm: 
    firstorder: 
    shape2D: 
    shape: 
    glrlm: 
    glszm: 
    gldm: 
    ngtdm:

### 2.4. Model Selection

The data were partitioned based on the chronological order of patient inclusion into training (n = 110), validation (n = 44), and test cohorts (n = 66) at the ratio of 5:2:3, for model training, validation, and independent evaluation, respectively. The endpoints were defined as binary outcomes: 6-month PFS, 12-month PFS, and 24-month OS. The supervised machine learning modeling was performed using the FeAture Explorer FAEv0.6.0.7z python package with NumPy, pandas, and scikit-learn [20]. The source code is openly available on GitHub (https://github.com/salan668/FAE.git (accessed on 25 May 2025)). The machine learning pipeline began with CSV files containing a binary outcome column and either CP-only, radiomics-only, or CP and radiomics feature columns. All features underwent identical normalization, feature selection, and classification procedures. Data balancing was performed through upsampling. At every step, preprocessing, feature selection, and classification, only one method was applied per each step, rather than combining multiple methods simultaneously. For example, feature pre-selection involved discarding features with Pearson’s correlation coefficient above 0.97 or, alternatively, applying principal component analysis. Feature selection was then performed using one of the following methods: ANOVA, Kruskal–Wallis (KW), Recursive Feature Elimination (RFE), or Relief. The remaining highly relevant features were used as input for one of these classifiers: support vector machine (SVM), linear discriminant analysis (LDA), logistic regression (LR), AdaBoost, Gaussian process (GP), multilayer perceptron (MLP), random forest (RF), least absolute shrinkage and selection operator (LASSO), decision tree (DT), or naïve Bayes (NB). The script was set to select between two and eight features. All possible combinations of the above components resulted in 1680 radiomics pipelines, trained on the development set and evaluated on the test dataset. Parameters such as slope, intercept, weight coefficients, and support vectors are learned from the data during training, while hyperparameters are not derived from the data. Instead, hyperparameters are tuned via grid-search based on the model’s performance on validation sets during cross-validation.

### 2.5. Ensemble Modeling

The probability scores delivered by each model were standardized to a mean of 0 and a standard deviation of 1 using z-score normalization. The top 15 prognostic models in the test set were then integrated, averaging their continuous normalized scores to generate an overall ensemble probability. These ensembles were subsequently also evaluated on the reserved test set. We tested varying numbers of models in the ensemble during preliminary experiments and found that including more than approximately 15 models no longer improved prognostic performance. “Soft” voting, which aggregates continuous probability scores, was used instead of hard voting binarized class labels to avoid biases introduced by early categorization.

### 2.6. Statistical Analysis

For each combination of features and endpoints, binary classification performance was evaluated on the reserved test set using the FAE Python package by calculating a comprehensive set of performance metrics, ROC analysis, F1 score, and the Youden index, to assess overall discrimination ability, while accuracy and balanced accuracy measure general correctness, with the latter accounting for class imbalance. True positives and true negatives were used as basic classification counts, while Matthews correlation coefficient (MCC) is a robust summary measure across all confusion matrix elements, especially in imbalanced datasets. Together, these metrics offer a wide and complementary evaluation of model performance.

### 2.7. Validation

Model performance was evaluated in two stages. First, ten-fold cross-validation was performed on the development set, using an internal validation subset within each fold. Next, the best-performing models in the test set were combined, and ensembles tested again on the test set, composed sequentially of the most recent 30% of patients. To prevent information leakage, the FAE radiomics pipeline was finalized prior to evaluation on the hold-out test subset.

## 3. Results

### 3.1. Patient Characteristics

We included 220 patients with inoperable NSCLC who received checkpoint inhibitors in a real-world setting (Table 1). The majority of patients were male (78.5%) and had stage IV disease (72.3%). Approximately half were smokers (46.4%) and had non-squamous histology (49.5%).

The vast majority of patients received either an atezolizumab-based combination (48.3%) or pembrolizumab (36.5%; Table 2). A response was achieved in 15.6% of patients. The median progression-free survival was 8.2 months [6.8–9.5], similar to that observed in registrational trials, and the median overall survival was 22.0 months (19.6–24.4, Table 2) [21].

### 3.2. Experimental Design

Figure 1 outlines the study workflow, which included CT scans from 220 patients. Briefly, patient and imaging data were curated, and 1680 CT-based radiomics models were trained, validated, and tested. These models were generated by combining three normalization methods, two dimensionality reduction techniques, four feature selection methods, and ten classifiers, with the number of selected features restricted to between two and eight (Figure 1). The split into development and test sets was based on the chronological order of patient inclusion, as detailed in the Methods section. This approach simulated a real-world clinical scenario in which retrospective data are used to predict outcomes for future patients from the same institution.

### 3.3. Performance of Individual Models

The best-performing individual models in the test set were then combined into an ensemble and evaluated in the reserved test set for stratification of PFS and OS following immune checkpoint inhibitor therapy. Models were developed for 6-, 12-, and 24-month endpoints, and the evaluation metrics included AUC, accuracy, balanced accuracy, true positives, true negatives, MCC, F1 score and the Youden index.

Among the top 15 individual models for the 24-month endpoint that included CP and radiomics features, the most frequent normalization methods were mean (40%) and min-max (33%). The most frequent preprocessing methods were PCC (53%) and PCA (47%). The most frequent feature selectors were Relief (60%) and Kruskal–Wallis (20%), while the most frequent classifiers were AdaBoost (60%) and Auto-Encoder (12%). The risk of insufficient diversification due to AdaBoost dominance is unlikely, because the three upstream optimization layers introduced additional variability in the selection of features entering AdaBoost. Figure 2 displays the classification evaluation of the individual models that best prognosticated 24-month overall survival in the test set. Applying a 24-month survival endpoint allowed us to define a patient subgroup whose PFS exceeded double that of the remaining cohort, highlighting those who derived maximal benefit from immunotherapy. As expected, individual models exhibited much higher prognostic performance on the training set compared to the validation and test sets (Figure 2). This performance gap reflects the degree of overfitting and serves as an indicator of each model’s generalizability.

We addressed potential temporal confounding by comparing prognostic performance in the chronologically selected test set with that in randomly selected validation subsets. This comparison provided a direct way to assess time-related bias. Among the 15 top-performing models for the 24-month endpoint that combined CP and radiomics features, the average AUC in the randomly selected validation subsets was 0.74 (SD 0.12, 95% CI 0.67 to 0.82), while, in the chronologically selected test set, the average AUC was 0.64 (SD 0.03, 95% CI 0.62 to 0.66). These AUC values refer to averages of individual models, not to the ensemble models. The AUC values obtained in the validation and test sets were significantly different, with t-statistic of 2.81 and *p* = 0.015 by an independent two-sample Welch t-test. Although the chronological design reduced performance, it remains the only clinically relevant internal modeling approach, because it mimics prospective prognostication within the same institution using retrospective data.

### 3.4. Performance of Ensemble Models

It is important to note that ensembles of the 15 best-performing models in the test set for each feature combination outperformed any individual model, except for the smallest model based solely on CP features (Figure 2 and Figure 3). This synergistic ensemble effect was most pronounced with the pooled CP and radiomics features for the 24-month endpoint, likely because the diversity of features provided synergistic benefits that greatly enhanced prognostic performance (Table 3 and Figure 3). In contrast to the individual models (Figure 2), the ensemble results (Table 3 and Figure 3) achieved superior prognostic performance when combining clinicopathological and radiomics features. Notably, the best-performing ensemble was obtained by using the 24-month endpoint (Table 3).

Figure 3 illustrates the good discrimination efficiency of the ensemble prognosticator which includes radiomics features and an excellent performance of the ensemble combining both CP and radiomics features. The continuous values of the CP and radiomics features prognosticator stratify a 100% homogenous group of non-survivors, comprising 42% of the total patients, without a single survivor (Figure 3).

## 4. Discussion

Computational analysis transforms clinical imaging into a rich source of quantitative features, gaining importance as imaging becomes more widely available and computational power continues to increase [4,22]. This study applied a machine learning-based radiomics approach to address the pressing clinical need for improved prediction of immunotherapy outcomes in NSCLC patients.

We hypothesized that the full prognostic potential of CT-derived tumor morphology has not been reached, largely due to limited experimentation within radiomics workflows, especially the underuse of ensemble modeling techniques. To address this gap, we implemented a stabilizing radiomics strategy that generates a diverse array of models and integrates them into a robust ensemble. This approach was aimed at enhancing predictive accuracy and minimizing the biases associated with individual models.

One of the key advantages of our study was the optimization of the radiomics modeling pipeline, resulting in the generation of 1680 prognostic models. This extensive exploration allowed the selection and ensembling of the top-performing models in the test set, thereby capturing complementary information across them, leading to improved performance. The ensemble models were then tested on the previously unobserved test dataset. This approach helps to reduce the bias associated with any individual model and to improve the classification performance in the test set [20,23]. In preliminary optimization, models using only one predictor underperformed those combining two or more features, while increasing the number of features beyond eight did not result in a notable improvement in performance Allowing models to include up to 30 features would have permitted more extensive pipeline exploration, producing approximately 10,000 candidate models; however, we prioritized robustness by restricting models to a small number of features for classifier construction. Other studies also used a small number of features for classifier construction to improve the generalizability [24].

Our best-performing ensemble model, which combined clinical parameters and radiomics features, achieved the AUC of 0.86 and C-index of 0.84 in predicting 24-month overall survival, surpassing most previously published results. This is likely attributable to the fact that only a few previous studies have utilized ensemble methods. However, the study by Upadhaya et al. reached AUCs up to 0.67 for predicting two-year overall survival in lung cancer patients, despite using an advanced ensemble methodology based on foundational artificial intelligence in addition to radiomics [8], while an ensemble deep learning strategy by Saad et al. achieved a C-index of 0.75 [25]. A deep learning ensemble model developed to predict recurrence of NSCLC at 12 months reached average AUCs up to 0.77 across validation folds, while test set results were not reported [26]. A similar predictive performance to our study was achieved by Gong et al., with AUCs of 0.89 and 0.85 in two validation cohorts employing a CT radiomics-based ensemble model [14]. However, its endpoint was not survival but brain metastasis occurrence [14]). Liu et al. employed a CT-based radiomics model to predict PFS and OS in NSCLC patients treated with nivolumab [27]. In their study, the average AUC values for predicting PFS and OS were 0.73 and 0.61, respectively. Multiple other efforts utilizing deep learning or logistic regression [5] have achieved AUCs ranging from 0.7 to 0.9 based on sample sizes between 48 and 1135 patients [7,28,29,30]. Improved performance by AUC of 0.823 was achieved by extraction of radiomics features from both primary tumors and lymph nodes using 18F-FDG PET imaging for predicting pathological complete response to neoadjuvant chemoimmunotherapy in NSCLC patients [6].

Similarly to the current study’s findings, Fried et al. combined pretreatment CT texture features with conventional prognostic features [31]. Models that integrated both textural features and conventional features demonstrated improved risk stratification for overall survival compared with models based solely on conventional features. In line with our current results, incorporating patient demographics and clinical factors alongside radiomics features has repeatedly been shown to enhance the predictive power of machine learning models [32]. Other studies reported that combining radiomics with clinical features only yielded AUCs up to 0.60 for predicting four-year progression-free and overall survival in NSCLC patients treated with Nivolumab or Pembrolizumab [15]. One study integrating clinical data and deep radiomics predicted survival in an independent test set after immune checkpoint inhibitor treatment by an AUCs of 0.824 and 0.753 for six- and nine-month survival, respectively [33].

One limitation of radiomics analysis is the difficulty in interpreting most features. Although many of our top-performing models included PCA-derived features, the best model that combined both clinicopathological and radiomics features did not rely on PCA, which allowed the easy interpretation of one of the four features in this model as the simple, intensity minimum of original images. In this case, a higher minimum intensity (resulting in a lighter image) is associated with a better outcome. PCA features are not interpretable because they are linear combinations of multiple original features. However, both PCA and image filter transformations (such as wavelet, gradient, logarithm) only marginally introduce additional opacity because most texture radiomics features are inherently abstract, being higher-order mathematical constructs, and often applied to transformed images. Therefore, radiomics accepts limited interpretability in favor of performance [34]. PCA-derived and native radiomics features were handled identically during feature selection and classification, ensuring no interpretability bias in the final models.

Another limitation of many radiomics studies is low reproducibility, due to a lack of standardization, insufficient reporting, or the absence of open-source code. We addressed this by using open-source code and providing the parameters file used for radiomics feature extraction. Additionally, we ensembled the best individual models to reduce variability in performance, which should further enhance reproducibility. Additionally, despite the objective nature of the computational analysis technique, the limitations of the workflow of this study included residual subjectivity due to the semi-automatic tumor VOI segmentation. Segmentation reproducibility by a single experienced radiologist. might have also impacted the model generalizability. Although intra- or interobserver variability was not assessed in this study, the semi-automatic 3D Slicer-based segmentation method used here has demonstrated high interobserver consistency in NSCLC [35]. Moreover, ensemble-based radiomics models with multistep feature selection have shown resilience to small segmentation variability [36].

Furthermore, radiomics studies are often based on retrospectively collected data and thus mainly serve as proof-of-concept, while prospective studies are needed to confirm the value of radiomics. To address this limitation, we divided the development and test sets in sequential chronological order, mirroring future routine clinical application of this methodology, in which the outcomes of current patients would be predicted using retrospective data.

## 5. Conclusions

We developed and evaluated multiple ensemble radiomics models to prognosticate OS and predict PFS in NSCLC by utilizing handcrafted radiomics signatures, clinical factors, and their combination. By designating the most recent CT scans as an internal test set, we simulated a realistic clinical scenario, whereby a model trained on retrospective data predicts outcomes for prospectively treated patients at the same institution. Our ensemble radiomics framework mitigates the limitations of single-model variability and enhances both generalizability and accuracy, thereby offering a template for more reproducible and accurate predictions of immunotherapy outcomes in future studies.

The achieved improvement in prognostic accuracy could enable personalized treatment by more reliable selection of NSCLC patients who are likely to benefit from immunotherapy. Further validation of pretreatment radiomics-based patient stratification should be pursued using larger and external datasets to confirm the possibility of a wider clinical applicability of this analytical workflow. Additionally, integrating radiomics with complementary modalities such as pathomics, genomics, proteomics, deep radiomics, and emerging markers like TMB and ctDNA could further enhance prognostic accuracy and improve treatment outcomes.

## Figures and Tables

**Figure 1 cancers-17-01790-f001:**
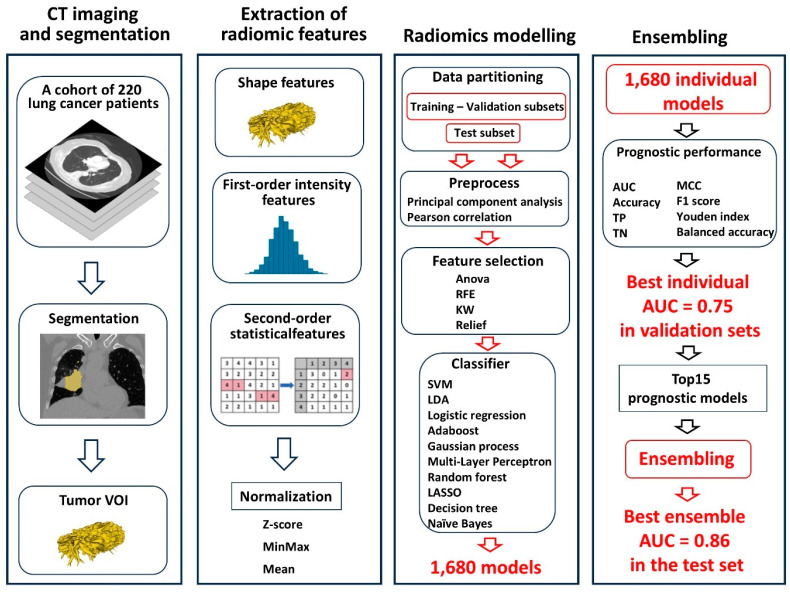
Flowchart of CT-based prognostic model construction for prediction of lung cancer PFS or prognostication of OS. CT scans were acquired after diagnosis, and tumor VOIs were segmented. Prior to radiomics analysis, images underwent normalization, resampling to 1 × 1 × 1 mm, and interpolation. A total of 2157 radiomics features were extracted using PyRadiomics. The CP and radiomics features were combined by adding them to a CSV file along with the outcome column. The cohort was further divided into development and test sets in a 70:30 ratio. Feature values were normalized using z-score, mean normalization (to −0.5, 0.5), or min-max normalization (to 0, 1). Subsequently, data balancing, preprocessing, feature selection, grid search for optimal hyperparameters, and classification were performed. Feature selection and classification were conducted by 10-fold cross-validation in the development set. The final prognostic model was constructed by combining the 15 best-performing models into ensembles using soft voting.

**Figure 2 cancers-17-01790-f002:**
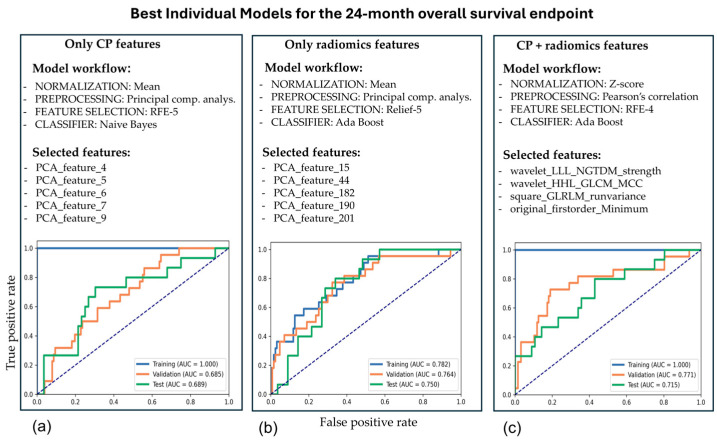
The classification performance of the best-performing individual models in the test set for the 24-month endpoint. The models were based on CP features alone, radiomics features alone, or a combination of CP and radiomics features. Panels (**a**–**c**) show the prognostic evaluation of the best-performing model in the test set for each feature combination.

**Figure 3 cancers-17-01790-f003:**
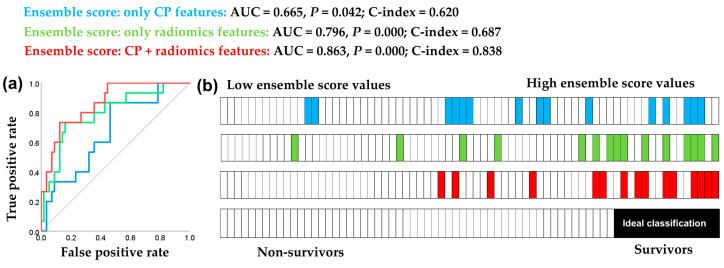
Classification performance of the ensembles for the 24-month endpoint using clinicopathological features alone, radiomics features alone, and a combined approach. The ensemble scores were calculated by summing up the probability scores of the 15 best-performing models. Panel (**a**) shows the ROC analysis of the three ensemble scores. Panel (**b**) provides a visual representation of the classification between non-survivors (white) and survivors (colored), with prognostic ensemble scores ordered from lowest (left) to highest (right) continuous values. It is obvious that as the ensemble scores increase, the likelihood of patient survival also increases.

**Table 1 cancers-17-01790-t001:** Clinical characteristics of patients included in the study ^a^.

	All Included (n = 220)	OS ≥ 24 Months (n = 52)	OS < 24 Months (n = 168)
**Sex**			
**Male**	172 (78.5%)	38 (73.1%)	135 (80.4%)
**Female**	48 (21.5%)	14 (26.9%)	33 (19.6%)
**Mean age (min-max)**	63.3 (35–87)	64.6 (38–82)	62.9 (35–87)
**Smoking**			
**Yes**	102 (46.4%)	31 (59.6%)	71 (42.3%)
**No**	118 (53.6%)	21 (40.4%)	97 (57.7%)
**NLR**			
**≥3**	109 (49.5%)	25 (48.1%)	84 (50%)
**Histology**			
**Adenocarcinoma**	109 (49.5%)	29 (55.8%)	80 (47.6%)
**Squamous cell cancer**	99 (45.0%)	21 (40.4%)	78 (46.4%)
**Large cell**	12 (5.5%)	2 (3.8%)	10 (6%)
**Stage**			
**IIIA**	33 (15.0%)	9 (17.3%)	24 (14.3%)
**IIIB**	28 (12.7%)	6 (11.5%)	22 (13.1%)
**IV**	159 (72.3%)	37 (71.2%)	122 (72.6%)
**Brain metastases**			
**Present**	16 (7.3%)	6 (11.5%)	10 (6.0%)
**Liver metastases**			
**Present**	17 (7.7%)	2 (3.8%)	14 (8.4%)

^a^ Tumor mutational burden (TMB) and circulating tumor DNA (ctDNA) biomarkers were not included due to incomplete availability across the cohort. Abbreviations: NLR, Neutrophil-to-Lymphocyte Ratio.

**Table 2 cancers-17-01790-t002:** Treatment characteristics and results.

	All Included (n = 220)	OS ≥ 24 Months (n = 52)	OS < 24 Months (n = 168)
**Drug**			
**Atezolizumab**	102 (48.3%)	19 (37.3%)	83 (51.9%)
**Pembrolizumab**	77 (36.5%)	20 (39.2%)	57 (35.6%)
**Nivolumab**	17 (8.1%)	3 (5.9%)	14 (8.8%)
**Prolgolimab**	15 (7.1%)	9 (17.6%)	6 (3.8%)
**Best response**			
**CR**	3 (1.4%)	0 (0%)	3 (1.8%)
**PR**	26 (11.8%)	12 (23.1%)	14 (8.3%)
**SD**	47 (21.4%)	9 (17.3%)	38 (22.6%)
**PD**	110 (50.0%)	27 (51.9%)	83 (49.4%)
**Not assessed**	34 (15.4%)	4 (7.7%)	30 (17.9%)
**RR**	29 (15.6%)	12 (25%)	17 (12.3%)
**DCR**	76 (40.8%)	21 (43.7%)	55 (39.9%)
**Median PFS (months)** **[95% CI]**	8.2 [6.8–9.5]	15.4 [11.9–18.9]	6.97 [6.2–7.8]
**Median OS (months)** **[95% CI]**	22.0 [19.6–24.4]	33.2 [30.7–35.7]	14.5 [12.3–16.7]

Abbreviations: CR, Complete Response; PR, Partial Response; SD Stable Disease; PD, Progressive Disease; RR, Response Rate; DCR; Disease Control Rate.

**Table 3 cancers-17-01790-t003:** Ensemble prognostic classification metrics were calculated on the reserved test set for models predicting 6-, 12-, and 24-month outcomes ^a,b,c^.

Features	AUC	AUC 95%CI	AUC *p*-Value	Accuracy	Balanced Accuracy	TP (%)	TN (%)	Low-Risk Count	High-Risk Count	MCC	F1 Score	Youden Index
**24-month overall survival**												
**CP**	0.671	0.525–0.818	0.043	0.76	0.60	41.7	83.1	59	12	0.23	0.37	0.20
**RADIOMICS**	0.796	0.666–0.927	0.000	0.82	0.79	55.0	92.2	51	20	0.52	0.62	0.57
**CP + RADIOMICS**	**0.863**	**0.769–0.957**	**0.000**	**0.85**	**0.80**	**61.1**	**92.5**	**53**	**18**	**0.57**	**0.66**	**0.61**
**12-month progression-free survival**												
**CP**	0.669	0.540–0.798	0.016	0.66	0.65	58.6	71.4	42	29	0.30	0.58	0.30
**RADIOMICS**	0.727	0.625–0.862	0.001	0.67	0.69	57.9	78.8	33	38	0.37	0.65	0.38
**CP + RADIOMICS**	**0.739**	**0.627–0.847**	**0.001**	**0.73**	**0.71**	**72.7**	**73.5**	**49**	**22**	**0.43**	**0.63**	**0.41**
**6-month progression-free survival**												
**CP**	0.675	0.542–0.812	0.015	0.73	0.63	78.6	53.3	15	56	0.29	0.822	0.26
**RADIOMICS**	0.701	0.565–0.837	0.009	0.72	0.64	79.2	50.0	18	53	0.28	0.81	0.27
**CP + RADIOMICS**	**0.719**	**0.550–0.828**	**0.013**	**0.76**	**0.71**	**84.2**	**57.1**	**21**	**50**	**0.42**	**0.83**	**0.42**

^a^ Prognostic evaluation was performed on the reserved internal test set, which comprised 30% of the patient cohort. ^b^ 1680 models were computed by combining three normalization methods, two preprocessing techniques, four feature selection methods, and ten classifiers. The models selected between two and eight features. ^c^ Bold formatting is used to highlight the best-performing model for each endpoint. Abbreviations: CP, clinical parameters; AUC, area under the curve; TP, true positives; TN, true negatives; MCC, Matthews Correlation Coefficient.

## Data Availability

The raw data supporting the conclusions of this article are available in Appendix A. The FAE source code is openly available on GitHub (https://github.com/salan668/FAE.git (accessed on 25 May 2025)).

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
