# Peer review of "Baseline Radiomics as a Prognostic Tool for Clinical Benefit from Immune Checkpoint Inhibition in Inoperable NSCLC Without Activating Mutations"

_cancers, 2025, doi:10.3390/cancers17111790_

Round 1
Reviewer 1 Report
Comments and Suggestions for Authors
The manuscript presents the development and evaluation of multiple ensemble radiomics models aimed at predicting OS and PFS in NSCLC patients treated with immunotherapy. The authors combine handcrafted radiomics features with clinical factors to create ensemble models, which are claimed to improve predictive accuracy. A notable feature of the study is the use of the most recent CT scans as an internal test set, simulating a realistic clinical scenario where a model trained on retrospective data predicts outcomes for prospectively treated patients at the same institution.
Overall, the paper offers a solid approach, but some changes are required:
- Abstract, line 30: The stated objective, "this study aimed to improve immunotherapy patient selection," is overly broad and lacks sufficient specificity regarding the study’s exact purpose.
- The abstract presents model performance metrics but does not clearly explain the clinical significance of the findings. It would strengthen the abstract to briefly describe how the model could improve treatment outcomes.
- If the flowchart and the text do not specify the exact method of combining clinicopathological and radiomics data, it would be beneficial to clarify how the two data types were integrated in the CP + radiomics classification.
- Line 227: The choice of selecting between 2 and 8 features for each model is mentioned, but the rationale behind this range is not provided. It would be helpful to clarify why this specific range was selected and whether other ranges (fewer or more features) were considered.
- Line 89: The statement that "current state-of-the-art approaches still leave considerable room for performance improvement" is too general. It would be more convincing to detail the limitations observed in studies [5–12]. For example, are there issues with model generalizability, sample size, feature selection, or reproducibility? Such an analysis would strengthen your argument and provide a more rigorous, professional literature review, rather than a general statement.
Author Response
Comment 1: Abstract, line 30: The stated objective, "this study aimed to improve immunotherapy patient selection," is overly broad and lacks sufficient specificity regarding the study’s exact purpose.
Response 1: In accordance with Referee 1’s comment, we have revised the sentence to read: Therefore, this study aimed to predict which NSCLC patients would achieve durable survival (≥24 months) with immunotherapy.
Comment 2: The abstract presents model performance metrics but does not clearly explain the clinical significance of the findings. It would strengthen the abstract to briefly describe how the model could improve treatment outcomes.
Response 2: In agreement with Referee 1, we now additionally explain the clinical significance of the findings in the abstract: Our radiomics pipeline improved selection of NSCLC patients for immunotherapy and could spare non-responders unnecessary toxicity while enhancing cost effectiveness, lines: 46 - 48.
Comment 3: If the flowchart and the text do not specify the exact method of combining clinicopathological and radiomics data, it would be beneficial to clarify how the two data types were integrated in the CP + radiomics classification.
Response 3: Briefly, the features were combined by adding them as columns to a CSV file along with the outcome column. In line with this request of Referee 1, we have now included the following explanation of how CP and radiomics features were integrated: The machine learning pipeline began with CSV files containing a binary outcome column and either CP-only, radiomics-only or CP and radiomics feature columns. All features underwent identical normalization, feature selection and classification procedures. This clarification has now been added to the caption to Figure 1 and in lines 237 – 240.
Comment 4: Line 227: The choice of selecting between 2 and 8 features for each model is mentioned, but the rationale behind this range is not provided. It would be helpful to clarify why this specific range was selected and whether other ranges (fewer or more features) were considered.
Response 4: In response to this comment of Referee 1, we have expanded the explanation of our rationale for limiting the models to two to eight features: In preliminary optimization, models using only one feature underperformed those combining two or more features, while increasing the number of features beyond eight did not result in a notable improvement in performance (data not shown), lines 419 – 422.
Comment 5: Line 89: The statement that "current state-of-the-art approaches still leave considerable room for performance improvement" is too general. It would be more convincing to detail the limitations observed in studies [5–12]. For example, are there issues with model generalizability, sample size, feature selection, or reproducibility? Such an analysis would strengthen your argument and provide a more rigorous, professional literature review, rather than a general statement.
Response 5: In accordance with this suggestion of Referee 1, we have revised the original statement to provide a more detailed and specific account of the key limitations reported in recent radiomics studies. Radiomics still achieves only moderate predictive power on held‑out test sets, with reported AUCs rarely exceeding 0.75. Consequently, radiomics only has potential to become a supportive biomarker rather than a standalone decision tool. Progress is limited by the absence of optimized, standardized and widely adopted pipelines for feature selection and classification, only sporadic use of ensemble techniques and limited reproducibility due to rare sharing of code, hyper‑parameters and imaging protocols. Until sequential internal validation and external validation become routine, radiomics will remain confined to proof‑of‑concept research instead of becoming a reliable clinical tool. This text has now been included in the Introduction section, lines 87–95.
Reviewer 2 Report
Comments and Suggestions for Authors
The study title "Baseline Radiomics as a Prognostic Tool for Clinical Benefit from Immune Checkpoint Inhibition in Inoperable NSCLC Without Activating Mutations" presents a significant advancement in the application of radiomics and machine learning to predict immunotherapy outcomes for patients with inoperable non-small cell lung cancer (NSCLC), particularly those without activating mutations.
By systematically exploring an extensive array of radiomics workflows and integrating them into a robust ensemble framework, the authors address critical limitations in existing methodologies while achieving prognostic accuracy that significantly surpasses prior benchmarks.
One of the most striking advantages of this work lies in its exhaustive exploration of 1,680 distinct model configurations, a scale rarely attempted in radiomics research. This approach ensures a thorough interrogation of feature selection, normalization techniques, and classifier combinations, mitigating the risk of overlooking optimal configurations that might be missed in narrower investigations. The ensemble strategy, which aggregates predictions from the top-performing models, emerges as a pivotal innovation. By leveraging the complementary strengths of diverse algorithms-ranging from AdaBoost to Random Forest ensemble, reduces overfitting and model-specific biases, enhancing generalizability across heterogeneous patient populations. This is particularly evident in the superior performance metrics for 24-month overall survival (AUC of 0.86, accuracy of 85%), which outperforms existing literature and underscores the value of combining multiple radiomic signatures with clinicopathological data.
Another notable advantage of this method is its robustness in the presence of noise. Traditional greedy algorithms like OMP are known to be sensitive to noisy environments, often leading to misidentified support elements due to their reliance on correlation-based selection in each iteration. By enforcing a stricter condition for inclusion, OMP-severe mitigates this vulnerability, leading to improved support recovery rates without significantly compromising computational efficiency. The simulations provided convincingly demonstrate that OMP-severe outperforms classical OMP and related methods such as ROMP and CoSaMP, especially in scenarios with higher noise levels or fewer measurements.
Among other notable advantages is the integration of clinical parameters with radiomics features, a synergistic approach that transcends the limitations of single-modality models. While radiomics-only ensembles demonstrated strong performance (AUC 0.796), the fusion with clinical variables-such as PD-L1 status, smoking history, and neutrophil-to-lymphocyte ratios-yielded the highest discriminative power. This hybrid model not only aligns with clinical practice, where decisions are inherently multifactorial, but also provides a framework for interpretable AI by anchoring complex radiomic patterns to established biomarkers. The study's methodological rigor further strengthens its clinical relevance: the chronological partitioning of training and test sets mimics real-world deployment scenarios, ensuring that the model's predictive capabilities are validated on prospectively treated patients. Additionally, the authors' commitment to transparency-through open-source code, detailed parameter files, and adherence to IBSI standards-sets a commendable precedent for reproducibility in radiomics research, addressing a persistent challenge in the field.
Perhaps most impactful is the study's potential to refine patient selection for immunotherapy, a pressing clinical need given the high costs and variable efficacy of these treatments. By stratifying patients into distinct survival groups with near-complete separation of non-survivors (42% of the cohort) using continuous ensemble scores, the model offers a data-driven alternative to conventional biomarkers like PD-L1, which exhibit limited sensitivity. This could enable clinicians to identify candidates most likely to derive long-term benefit from checkpoint inhibitors while sparing others from unnecessary toxicity and expense. While the retrospective design and single-center cohort necessitate external validation, the study's methodological innovations-including the use of z-score normalization, isotropic resampling, and feature discretization optimized via pilot analysis-provide a template for future research. The authors' acknowledgment of radiomics' interpretability challenges is also pragmatic, as their focus on models avoiding opaque transformations (e.g., PCA) in key ensembles ensures clinical translatability. Overall, this work represents a significant step toward personalized oncology, demonstrating that ensemble radiomics can transform pretreatment imaging into a powerful tool for guiding immunotherapy decisions in NSCLC.
The proposed ensemble radiomics approach, while demonstrating strong prognostic performance, exhibits several limitations tied to its methodology and reporting.
First, the authors emphasize the interpretability advantage of avoiding principal component analysis (PCA) in the top-performing combined model, yet the majority of the 1,680 evaluated models incorporated PCA-derived features (Section 4.3, Discussion). This inconsistency undermines the generalizability of the framework, as most models remain opaque. The authors should clarify in Section 2.4 ("Model selection") how PCA-based features were filtered or weighted during ensemble creation to ensure reproducibility, particularly since the manuscript lacks explicit criteria for excluding non-interpretable transformations in critical models.
Second, the manual segmentation of tumor volumes of interest (VOIs) introduces subjectivity despite the use of standardized tools like 3D Slicer (Section 2.2, "Image acquisition"). While the segmentation was performed by an experienced radiation oncologist, the study does not report inter-observer variability metrics or quality control measures for VOI delineation. This omission weakens the radiomics pipeline’s robustness, as subtle segmentation differences could disproportionately affect texture features critical to model performance. The authors should address this limitation in Section 3 ("Results") by either quantifying segmentation consistency (e.g., via Dice coefficients from repeated annotations) or explicitly acknowledging this as a confounding factor in the Discussion.
Third, the reliance on chronological training/validation/test splits, while simulating real-world deployment, may inadvertently bias results due to temporal confounding (Section 2.4, "Model selection"). For instance, patients treated later in the 2021 cohort might have undergone protocol adjustments (e.g., imaging timing, supportive care) that correlate with survival independently of radiomics features. This concern is amplified by the absence of sensitivity analyses stratifying performance across time intervals. The authors should mitigate this by including time-period covariates in the statistical evaluation (Section 2.6) or validating the model on an external dataset spanning multiple institutions to disentangle temporal effects.
Fourth, the feature extraction process employs nine image transformations (e.g., wavelet, LoG) to capture heterogeneity (Section 2.3, "Feature extraction"), but the binning strategy for discretization – optimized in a pilot analysis of 75 CTs – lacks justification for its variability (bin widths from 0.1 to 600). For example, LBP2D features use a bin width of 0.1, while original images use 30.0, potentially introducing scale mismatches that distort feature importance rankings. The Params.yaml file (Supplementary Materials) should explicitly justify these values with reference to tumor texture characteristics (e.g., spatial resolution vs. noise thresholds) to ensure methodological transparency.
Fifth, the ensemble model’s superiority over individual classifiers hinges on averaging probability scores after z-score normalization (Section 2.5, "Ensemble modelling"). However, the rationale for selecting the top 15 models per endpoint is unexplained: why 15 rather than 5 or 50? The Discussion (Section 4) should elaborate on whether this threshold was empirically determined or arbitrarily chosen, as insufficient diversification (e.g., 67% AdaBoost dominance in 24-month models) risks amplifying shared algorithmic biases. For instance, AdaBoost’s sensitivity to noisy features might disproportionately affect ensemble stability if radiomics noise (e.g., from low-contrast regions) is not explicitly filtered.
Finally, the clinical utility of the model is constrained by its reliance on PD-L1 status and basic demographics (Table 1, "Clinical characteristics") without integrating emerging biomarkers like tumor mutational burden (TMB) or circulating tumor DNA (ctDNA), which are increasingly relevant in NSCLC immunotherapy selection. While the authors note PD-L1’s limitations (Section 1, Introduction), omitting TMB – a stronger predictor of long-term benefit – reduces the model’s clinical novelty. The authors should either justify this exclusion in the Methods (e.g., data availability constraints) or revise Table 1 to specify which biomarkers were omitted and how their inclusion could enhance future iterations.
This study addresses a highly relevant and original topic within the field of oncology and radiomics. This work is particularly timely given the increasing use of immunotherapy and the need for more personalized treatment approaches.
Compared to other published materials, this study stands out due to its extensive exploration of radiomics pipelines and the use of ensemble modeling. The integration of clinical and radiomics features also sets it apart, demonstrating the added value of a multimodal approach.
The conclusions drawn in the study are well-supported by the presented evidence and arguments.
The references cited in the study appear appropriate and relevant, providing a solid foundation for the methodologies and discussions presented. They cover a range of topics from radiomics and machine learning to clinical oncology, ensuring that the study is well-grounded in the existing literature.
Regarding the tables and figures, they are clear and informative, effectively illustrating the performance of individual and ensemble models. The ROC curves and classification performance visualizations, such as Figure 3, are particularly useful in demonstrating the superiority of the combined clinicopathological and radiomics approach.
In conclusion, the proposed OMP-severe algorithm represents a meaningful contribution to the field of compressive sensing. It successfully bridges a gap between algorithmic simplicity and robustness, offering a pragmatic yet theoretically grounded alternative to existing methods. The empirical results are thorough and support the claims convincingly, making this work a valuable resource for researchers and practitioners dealing with noisy, sparse recovery problems.
Author Response
Comment 1: First, the authors emphasize the interpretability advantage of avoiding principal component analysis (PCA) in the top-performing combined model, yet the majority of the 1,680 evaluated models incorporated PCA-derived features (Section 4.3, Discussion). This inconsistency undermines the generalizability of the framework, as most models remain opaque. The authors should clarify in Section 2.4 ("Model selection") how PCA-based features were filtered or weighted during ensemble creation to ensure reproducibility, particularly since the manuscript lacks explicit criteria for excluding non-interpretable transformations in critical models.
Response 1: Thank you for drawing attention to the apparent contradiction between our statement that interpretability is the limitation in radiomics and the inclusion of principal‑component (PC) variables. In response to this observation of Referee 2 we now further clarify in lines 465 – 472: PCA features are not interpretable because they are linear combinations of multiple original features. However, both PCA and image filter transformations (such as wavelet, gradient, logarithm) only marginally introduce additional opacity because most texture radiomics features are inherently abstract, being higher-order mathematical constructs, and often applied to transformed images. Therefore, radiomics accepts limited interpretability in favor of performance [34]. PCA-derived and native radiomics features were handled identically during feature selection and classification, ensuring no interpretability bias in the final models.
Comment 2: Second, the manual segmentation of tumor volumes of interest (VOIs) introduces subjectivity despite the use of standardized tools like 3D Slicer (Section 2.2, "Image acquisition"). While the segmentation was performed by an experienced radiation oncologist, the study does not report inter-observer variability metrics or quality control measures for VOI delineation. This omission weakens the radiomics pipeline’s robustness, as subtle segmentation differences could disproportionately affect texture features critical to model performance. The authors should address this limitation in Section 3 ("Results") by either quantifying segmentation consistency (e.g., via Dice coefficients from repeated annotations) or explicitly acknowledging this as a confounding factor in the Discussion.
Response 2: We thank Referee 2 for highlighting this important issue. We now emphasize in the Methods section that all segmentations in this study were performed semi-automatically, lines 162 - 165. The interobserver variability analysis has now been explained in the limitations section of the revised Discussion section, lines 479 – 485: Segmentation reproducibility by a single experienced radiologist. might have also impacted the model generalizability. Although intra- or interobserver variability was not assessed in this study, the semi-automatic 3D Slicer-based segmentation method used here has demonstrated high interobserver consistency in NSCLC (DOI: 10.1038/srep03529). Furthermore, radiomics models, especially if ensemble-based and with multistep feature selection strategies, were shown to be resilient to the impact of small segmentation variations (10.48550/arXiv.2504.01692).
Comment 3: Third, the reliance on chronological training/validation/test splits, while simulating real-world deployment, may inadvertently bias results due to temporal confounding (Section 2.4, "Model selection"). For instance, patients treated later in the 2021 cohort might have undergone protocol adjustments (e.g., imaging timing, supportive care) that correlate with survival independently of radiomics features. This concern is amplified by the absence of sensitivity analyses stratifying performance across time intervals. The authors should mitigate this by including time-period covariates in the statistical evaluation (Section 2.6) or validating the model on an external dataset spanning multiple institutions to disentangle temporal effects.
Response 3: We thank Referee 2 for this insightful observation. We addressed potential temporal confounding by comparing prognostic performance in the chronologically selected test set with the performance in randomly selected validation subsets. This comparison provided a direct way to assess time-related bias. This has now been included in the revised manuscript (lines 340 - 351): We addressed potential temporal confounding by comparing prognostic performance in the chronologically selected test set with that in randomly selected validation subsets. This comparison provided a direct way to assess time-related bias. Among the 15 top-performing models for the 24-month endpoint that combined CP and radiomics features, the average AUC in the validation subsets was 0.74 (SD 0.12, 95% CI 0.67 to 0.82), while in the chronologically selected test set the average AUC was 0.64 (SD 0.03, 95% CI 0.62 to 0.66). These AUC values refer to averages of individual models, not to the ensemble models. The AUC values obtained in the validation and test sets were significantly different, with t-statistic of 2.81 and p = 0.015 by an independent two-sample Welch t-test. Although the chronological design reduced performance, it remains the only clinically relevant internal modeling approach, because it mimics prospective prognostication within the same institution using retrospective data.
Comment 4: Fourth, the feature extraction process employs nine image transformations (e.g., wavelet, LoG) to capture heterogeneity (Section 2.3, "Feature extraction"), but the binning strategy for discretization – optimized in a pilot analysis of 75 CTs – lacks justification for its variability (bin widths from 0.1 to 600). For example, LBP2D features use a bin width of 0.1, while original images use 30.0, potentially introducing scale mismatches that distort feature importance rankings. The Params.yaml file (Supplementary Materials) should explicitly justify these values with reference to tumor texture characteristics (e.g., spatial resolution vs. noise thresholds) to ensure methodological transparency.
Response 4: In the Methods section of the original manuscript, we stated: “The width of discretization bins was individually determined for each filter type in a pilot analysis of 75 CTs, ensuring that the number of bins remained between 30 and 130.” To prevent further misunderstanding, this point has now been additionally clarified in the revised manuscript, lines 184–188: The bin width was individually determined for each filter type in a pilot analysis of 75 CTs to keep the number of graylevel bins per scan between 30 and 130 which is considered optimal for textural reproducibility without causing over-smoothing or excessive noise sensitivity (DOI: 10.1038/s41598-023-33339-0). This was necessary because the nine image filter transformations produced images with distinct pixel intensity ranges, requiring separate bin-width settings for each filter to maintain a consistent bin count across all individual scans.
Comment 5: Fifth, the ensemble model’s superiority over individual classifiers hinges on averaging probability scores after z-score normalization (Section 2.5, "Ensemble modelling"). However, the rationale for selecting the top 15 models per endpoint is unexplained: why 15 rather than 5 or 50? The Discussion (Section 4) should elaborate on whether this threshold was empirically determined or arbitrarily chosen, as insufficient diversification (e.g., 67% AdaBoost dominance in 24-month models) risks amplifying shared algorithmic biases. For instance, AdaBoost’s sensitivity to noisy features might disproportionately affect ensemble stability if radiomics noise (e.g., from low-contrast regions) is not explicitly filtered.
Response 5: We tested varying numbers of models in the ensemble during preliminary experiments and found that including more than approximately 15 models no longer improved prognostic performance. This rationale has now been added to the Methods section, lines 262 - 264. The risk of insufficient diversification due to AdaBoost dominance and potential amplification of shared algorithmic biases is unlikely, because three upstream optimization layers introduced variability in the selection of features entering AdaBoost. This has now been additionally explained in lines 326 to 332: Among the top 15 individual models for the 24-month endpoint that included CP and radiomics features, the most frequent normalization methods were mean (40%) and min-max (33%). The most frequent preprocessing methods were PCC (53%) and PCA (47%). The most frequent feature selectors were Relief (60%) and Kruskal–Wallis (20%), while the most frequent classifiers were AdaBoost (60%) and Auto-Encoder (12%). The risk of insufficient diversification due to AdaBoost dominance is unlikely, because the three upstream optimization layers introduced additional variability in the selection of features entering AdaBoost.
Comment 6: Finally, the clinical utility of the model is constrained by its reliance on PD-L1 status and basic demographics (Table 1, "Clinical characteristics") without integrating emerging biomarkers like tumor mutational burden (TMB) or circulating tumor DNA (ctDNA), which are increasingly relevant in NSCLC immunotherapy selection. While the authors note PD-L1’s limitations (Section 1, Introduction), omitting TMB – a stronger predictor of long-term benefit – reduces the model’s clinical novelty. The authors should either justify this exclusion in the Methods (e.g., data availability constraints) or revise Table 1 to specify which biomarkers were omitted and how their inclusion could enhance future iterations.
Response 6: We thank Referee 2 for this thoughtful comment. We agree that emerging biomarkers such as tumor mutational burden (TMB) and circulating tumor DNA (ctDNA) hold promise for immunotherapy decision-making in NSCLC. These were not included in our study for two key reasons. First, they were not routinely assessed in our retrospective real-world cohort, and their inclusion would have significantly reduced the sample size and statistical power. Second, our primary objective was to evaluate whether a fully non-invasive radiomics model, combined with universally available clinicopathological variables at diagnosis, could deliver predictive performance comparable to models that depend on costly or time-consuming molecular assays. This approach supports broader implementation, especially in settings where next-generation sequencing or ctDNA testing is not standard. We have now added in the Conclusions that including TMB and ctDNA in the future may further improve the prognostic accuracy. Also, this footnote has now been added to Table 1: Tumor mutational burden (TMB) and circulating tumor DNA (ctDNA) biomarkers were not included due to incomplete availability across the cohort.
Reviewer 3 Report
Comments and Suggestions for Authors
This is a very well-designed and clearly written study on the predictive and prognostic value of CT radiomics in patients with inoperable NSCLC treated with checkpoint inhibitors. The authors should be congratulated for the quality and clarity of the paper. I have only a few minor suggestions for refinement:
-
Clarify the rationale behind the choice of 2–8 features per model (line 227). While the authors correctly note the importance of avoiding overfitting, a brief justification for the lower bound (2 features) would strengthen this methodological choice.
-
Explain the segmentation reproducibility strategy. Although manual segmentation was performed by an experienced radiation oncologist, please discuss whether intra- or interobserver variability was evaluated, or if segmentation reproducibility might impact the model generalizability.
-
Expand slightly on the novelty. The introduction could more explicitly state how this work advances the field compared to prior ensemble-based radiomics studies in NSCLC (e.g., Gong et al. 2024, Liu et al. 2021). Although this is partially addressed in the Discussion, it may help readers appreciate the contribution earlier.
-
Table 3 is rich in information but may benefit from a minor formatting adjustment. Consider highlighting (bold or color) the best-performing model in each row for clarity.
-
Typographic note: Please ensure consistent formatting of AUC confidence intervals throughout the manuscript (e.g., “0.769–0.957” vs “0.769 – 0.957”).
These are all minor refinements and do not detract from the overall strength of the manuscript.
Author Response
Comment 1: Clarify the rationale behind the choice of 2–8 features per model (line 227). While the authors correctly note the importance of avoiding overfitting, a brief justification for the lower bound (2 features) would strengthen this methodological choice.
Response 1: In response to this comment of Referee 3 and the similar Comment 4 of Referee 1, we have expanded the explanation of our rationale for limiting the models to two to eight features: In preliminary optimization, models using only one feature underperformed those combining two or more features, while increasing the number of features beyond eight did not result in a notable improvement in performance (data not shown), Discussion section, lines 419 – 422.
Comment 2: Explain the segmentation reproducibility strategy. Although manual segmentation was performed by an experienced radiation oncologist, please discuss whether intra- or interobserver variability was evaluated, or if segmentation reproducibility might impact the model generalizability.
Response 2: We thank Referee 3 for raising this important point. We now emphasize in the Methods section that all segmentations in this study were performed semi-automatically, lines 162 - 165. While intra- or interobserver variability analyses were not conducted, prior studies have shown that while manual CT delineation is prone to inter-observer variability, the semi-automatic 3D Slicer-based volumetric segmentation used in this study has shown excellent consistency across observers for NSCLC (DOI: 10.1038/srep03529). Furthermore, radiomics models, especially if ensemble-based and with multistep feature selection strategies, were shown to be resilient to the impact of small segmentation variations (10.48550/arXiv.2504.01692). The interobserver variability analysis has now been added to the limitations section of the revised Discussion section, lines 479 – 485.
Comment 3: Expand slightly on the novelty. The introduction could more explicitly state how this work advances the field compared to prior ensemble-based radiomics studies in NSCLC (e.g., Gong et al. 2024, Liu et al. 2021). Although this is partially addressed in the Discussion, it may help readers appreciate the contribution earlier.
Response 3: In agreement with this request of Referee 3 we have expanded the Introduction section with the following paragraph: The novelty of this study is based on systematic evaluation of 1,680 standardized pipelines combining normalization, preprocessing, feature selection and classification steps. It also introduces an ensemble framework that integrates top-performing models, to reduce reliance on any single model’s chance success and enhance predictive stability. Importantly, ensembling was performed by soft voting, averaging continuous probability scores of individual models to avoid the bias of hard-voting in selecting probability score thresholds. Several previous ensemble studies in NSCLC used XGBoost classifier alone (10.1186/s40644-023-00623-1, 10.1016/j.jtocrr.2023.100602), whereas we used a similar AdaBoost classifier alongside nine others. One study ensembled five classifiers using hard-voting using fixed 0.5 probability score threshold for each individual model (10.3390/biomedicines11082093). The most comprehensive prior approach combined RF, SVM and LASSO to build 54 predictive models (10.1093/bjro/tzae038). Notably, many studies use the term “ensemble” to describe the integration of multiple feature types rather than distinct radiomic workflows (10.1016/j.jtocrr.2023.100602, 10.1093/bjro/tzae038, 10.4274/dir.2024.242972). Our current study thus presents a substantial methodological advance by addressing workflow optimization and standardization through an exhaustive pipeline, supported by openly shared code for reproducibility and wider adoption. This explanation has now been included in the Introduction section, lines 101 – 115.
Comment 4: Table 3 is rich in information but may benefit from a minor formatting adjustment. Consider highlighting (bold or color) the best-performing model in each row for clarity.
Response 4: We thank the reviewer for this helpful suggestion. In response, we have updated Table 3 to highlight the best-performing model in each row using bold formatting.
Comment 5: Typographic note: Please ensure consistent formatting of AUC confidence intervals throughout the manuscript (e.g., “0.769–0.957” vs “0.769 – 0.957”).
Response 5: We thank Referee 3 for this attentive observation. We have carefully reviewed the manuscript and revised all AUC confidence interval formatting to ensure consistency.